# Body Temperature Monitoring for Regular COVID-19 Prevention Based on Human Daily Activity Recognition

**DOI:** 10.3390/s21227540

**Published:** 2021-11-12

**Authors:** Lei Zhang, Yanjin Zhu, Mingliang Jiang, Yuchen Wu, Kailian Deng, Qin Ni

**Affiliations:** 1College of Information Science and Technology, Donghua University, Shanghai 201620, China; lei.zhang@dhu.edu.cn (L.Z.); 2211828@dhu.edu.cn (Y.Z.); 170910829@dhu.edu.cn (M.J.); wyc15180543539@163.com (Y.W.); dengkailian@dhu.edu.cn (K.D.); 2College of Information, Mechanical and Electrical Engineering, Shanghai Normal University, Shanghai 201418, China

**Keywords:** human activity recognition (HAR), wearable sensors, COVID-19, temperature sensor, machine learning (ML)

## Abstract

Existing wearable systems that use G-sensors to identify daily activities have been widely applied for medical, sports and military applications, while body temperature as an obvious physical characteristic that has rarely been considered in the system design and relative applications of HAR. In the context of the normalization of COVID-19, the prevention and control of the epidemic has become a top priority. Temperature monitoring plays an important role in the preliminary screening of the population for fever. Therefore, this paper proposes a wearable device embedded with inertial and temperature sensors that is used to apply human behavior recognition (HAR) to body surface temperature detection for body temperature monitoring and adjustment by evaluating recognition algorithms. The sensing system consists of an STM 32-based microcontroller, a 6-axis (accelerometer and gyroscope) sensor, and a temperature sensor to capture the original data from 10 individual participants under 4 different daily activity scenarios. Then, the collected raw data are pre-processed by signal standardization, data stacking and resampling. For HAR, several machine learning (ML) and deep learning (DL) algorithms are implemented to classify the activities. To compare the performance of different classifiers on the seven-dimensional dataset with temperature sensing signals, evaluation metrics and the algorithm running time are considered, and random forest (RF) is found to be the best-performing classifier with 88.78% recognition accuracy, which is higher than the case of the absence of temperature data (<78%). In addition, the experimental results show that participants’ body surface temperature in dynamic activities was lower compared to sitting, which can be associated with the possible missing fever population due to temperature deviations in COVID-19 prevention. According to different individual activities, epidemic prevention workers are supposed to infer the corresponding standard normal body temperature of a patient by referring to the specific values of the mean expectation and variance in the normal distribution curve provided in this paper.

## 1. Introduction

Following the popularization and optimization of intelligent electronic devices and computer systems, human activity recognition (HAR) via wearable devices has been attracting research attention for decades. HAR technology has been applied to various scenarios in the fields of medical care, military security, health service and infotainment, and new application requirements have even emerged in the context of COVID-19 [1,2,3,4,5,6,7,8]. In comparison with vision-based activity recognition, wearable sensors in human activity recognition provide more possibilities for human and mobile devices to interact due to their low cost, small size, strong computing ability and privacy protection, which also serve as the hardware support for the experimental data collection stage. Tools such as machine learning (ML) and deep learning (DL) act as core learning algorithms, allowing raw data on human activity recognition to be generalized in various domains after training and testing. Wearable sensing devices typically combine embedded systems with inertial, physiological or environmental sensors to identify ambulation, exercise and daily activities, thus enabling HAR technology not only to perform monitoring and activity prediction but also to provide personalized service and decision support under certain circumstances [4,9,10,11,12,13,14,15,16,17].

### 1.1. Related Work

HAR technology using wearable devices and learning algorithms has already achieved several successful applications. Yen et al. proposed an automatic CNN feature extraction method combined with a wearable inertial sensor to improve the accuracy of daily activity recognition [4]. Lawal and BanoI developed a method of establishing a frequency image for an original signal to improve the overall recognition performance [10]. N.K. et al. proposed a collaborative optimization technology for a sensor classifier based on HAR, which can also reduce energy consumption [11]. As the scalar output and pool in CNN cannot be equivalent, Pham et al. proposed a wearable sensor using the capsule network to ensure activity recognition [12]. Khokhlov et al. assumed that smart phones with embedded accelerometers and gyroscope sensors can be used to guarantee accuracy in activity recognition devices [14]. Hsu et al. presents a wearable inertial sensor network that can effectively identify 11 kinds of sports activities; however, its results are still affected by various activity datasets, and the resource consumption in the process of data transmission and processing is also high [9]. Ayman et al. established a new HAR framework based on the IMU sensor, which forms an efficient HAR system [15]. Wu et al. proposed a wireless vital signs monitoring system, which can be applied to daily medical care applications [16]. He et al. proposed a new HAR model based on cyclic attention learning, which is superior to the traditional CNN and LSTM models, and it also shows how to recognize special human activities on weak-labeled sensor datasets through end-to-end training and achieves satisfying results [17].

Since the outbreak of COVID-19 around the world, many new application demands and values based on wearable sensing technology have also been constantly emerging. A number of similar related works have been proposed in the past two years. Ueafuea et al. reviewed the application of mobile and wearable devices in providing psychological support against the background of the COVID-19, which is primarily aimed at front-line workers with psychosocial diseases [5]. Lonini et al. proposed the use of flexible wearable sensing devices and structured activities to quickly screen the physiological changes of patients with novel coronavirus [6]. Sadighbayan and Ghafar summarized portable sensing devices for COVID-19 [7]. A contactless, small-scale motion monitoring system using software-defined radio was designed by Rehman et al. for the early diagnosis of COVID-19 [17]. However, the quality and efficiency of the follow-up long-term prevention and control efforts will be of greater concern as the epidemic begins to stabilize and be brought under control in most countries and regions. Nevertheless, as the infectious disease is currently under control in some countries and regions, more efforts should be placed on improving the quality and efficiency of the subsequent long-term epidemic prevention work.

Strict screening and timely isolation can effectively respond to epidemic prevention and control. Fever is one of the prominent clinical symptoms of patients with novel coronavirus, so body temperature monitoring is a key step of daily epidemic prevention, and infrared thermometers play an important role as a common temperature measurement tool. Generally speaking, when a person’s body temperature exceeds 37.3, it means that they have reached the fever standard. An infrared thermometer is able to maintain a certain distance for temperature measurement and avoid large-scale population retention and cross-infection, thereby reducing the risk of contact and transmission. Therefore, it is suitable for the preliminary screening of crowds, including public places such as campuses, communities, supermarkets, office buildings and transportation hubs. Figure 1 illustrates some sample pictures of temperature measurement in some domestic and overseas public places. However, the accuracy of temperature recordings cannot always be guaranteed in some cases. On one hand, data are easily affected by the surrounding environment, such as the ambient temperature and humidity, because the measured part is usually exposed. The best measurement range of the thermometer is 16–35 ${}^{{}^{\circ}}$C, while the outdoor temperature in winter obviously cannot reach this level. On the other hand, the experimental part mentioned in the following section reveals that the human body temperature under a motion state will be lower than that under a stationary state when the environmental temperature remains constant. Consequently, extreme weather conditions and human activity states influence the accuracy of temperature recordings and also affect the efficiency of body temperature monitoring.

### 1.2. Main Contributions

On the basis of the current social situation and compared to existing related work, our main contributions are as follows:In addition to the accelerometer and gyroscope, the independently designed wearable device also adds the temperature sensor module, which enriches the sensing data and is conducive to the further study of the relationship between human body surface temperature and the accuracy of activity recognition.The 10 participants spanned the major age groups, various professions and different ranges of height and weight. After collecting sensing data under similar experimental conditions, the data are divided into a training set (75%) and testing set (25%) for learning algorithms to ensure that the selected learning model is generalized enough to adapt better to future new users.The performance of almost all algorithms has been improved to varying degrees after incorporating body surface temperature data (slightly lower than the normal human body temperature). In other words, the temperature sensing data achieve more accurate human activity recognition.Among all the selected learning modules, random forest (RF) and extreme trees (ET) comprehensively perform better. Without data stacking, the ET reaches an 89% recognition rate and RF reaches an 88% recognition rate, while it has less computing time consumption. After the resampling process, the performances of the algorithms with the raw dataset are continuously improved, and ET and RF can reach 90% and 92% accuracy, respectively.The body surface temperature of participants under moving activities (walking, walking upstairs or downstairs) is lower than that of sitting, which is related to the body temperature monitoring during COVID-19. Temperature errors of 1–2 ${}^{{}^{\circ}}$C may lead to the omission of potential feverous people and affect the accuracy and efficiency of epidemic prevention work.

The remainder of this paper is organized as follows: Section 2 presents the experimental setup process, including hardware device components, participants’ demographics and some visualized data examples. Section 3 introduces signal standardization and data stacking and the activity recognition algorithms’ accuracy, as well as the model evaluation results. The influence of human surface temperature data on the algorithms’ performance is also attached great importance in this section. Section 4 links practical human-centered applications in the context of COVID-19. Finally, Section 5 provides the conclusions and future work.

## 2. Experimental Setup

### 2.1. Apparatus

The wearable hardware device employed in this experiment is composed of a STM32 based microcomputer, an inertial sensor module (including a three-axis accelerometer and a three-axis gyroscope), a temperature sensor, a power supply and a Bluetooth module used to realize the wireless communication between the device and computer (Figure 2). A detailed description of each component of the hardware support is summarized in Table 1.

The size of the microcontroller (STM32F103C8T6) is 64 mm × 36.4 mm, comprising a STM32F 103C8 processor, a LM1117-3.3V voltage regulator chip, a miniUSB interface used for system power and a reserved serial port. The inertial sensor (MPU6050) consists of a triaxial accelerometer with a range of ±2 g and a triaxial gyroscope with a range of ±2000 dps, which can simultaneously collect the human activity data. The sampling rate of the sensing system in the experiment was set at 50 Hz and converted into a digital signal output to synthesize the three-dimensional activity space of acceleration and angular velocity. LMT70 supports STM32 MCU, which was used for contact temperature measurement. The bottom plate of Bluetooth HC-06 is equipped with a Bluetooth core module, with six rows of needles directly connected to the serial port of SCM, and the effective communication distance of HC-06 is within 10 m. The power supply module of the whole device is a 5 V, 1500 mAh dry battery.

The above hardware structure was designed for this experiment for several reasons. Firstly, the STM32F103C8T6 microcontroller has a relatively small volume and uses J-Link for procedures, and its chip performance and peripheral configuration are suitable for the working of the main control chip. The inertial module was used to measure x, y and z under different states of three-axis acceleration, as well as three-axis angular velocity pitch, roll and yaw. The temperature sensor LMT70 must be close to the skin to obtain accurate data because it is a contact chip. In addition, the wearable device communicates with the computer through Bluetooth, and the sensor signal is sent to the serial port through Bluetooth module. The computer uses Python to read the serial port data, and then stores the data in Excel to prepare for the subsequent data analysis and processing step. After establishing and matching the above modules, the wearable device lays a foundation for supporting data collection and visualization analysis procedures.

### 2.2. Participants

This study covered 10 healthy participants (5 males and 5 females), and Table 2 presents their body parameter information (age, height and weight). Each subject was assigned an ID number to protect their private information and was informed of the experimental processes before the experiment started. After the measurement of body parameters at the appointed time and place, the participants were required to wear wearable sensing devices on the same parts. During the data collection process, the wearable sensor needed to be worn on the wrist of the experimental participant to ensure accurate data transmission to reduce deviations, and the temperature sensor was attached to the inside of the wrist with a bandage in order to close to the skin. Next, the power supply and Bluetooth serial port of the sensor were opened to start the formal data collection experiment, and the edited Python code was compiled after successfully connecting to the computer. Participants completed four activities (sitting, walking, walking upstairs and walking downstairs) successively in a natural way (Figure 3).

The seven-dimensional serial port data was read and stored in the cloud, including three-axis acceleration, three-axis angular velocity and body surface temperature. Since the major signal was the directional data collected by the inertial sensor, the whole sensor needed to be stabilized with an elastic bandage during the experiment. The acquisition frequency was constant at 50 Hz, and the direction and position of the device needed to be ensured to avoid inaccurate data caused by the inertial sensor placement. The HAR system was run according to pre-designed procedures, and the program collected data at a speed of about 15 data points within 1 s, 50–60 s per cycle, and stored the data according to different activities in a local Excel file. The data collection process was not completed until four different activities of all participants were successfully collected. In the end, a total of 31,713 entries were obtained, including 8365 tags for sitting, 9416 tags for walking, 6581 tags for walking upstairs and 7351 tags for walking downstairs, which were input as the raw data set before data preprocessing.

### 2.3. Activity Data Visualization

In order to observe and analyze the differences of the data more intuitively for the four activities, the signal diagrams from seven dimensions were visualized by using the Matplotlib module in Python before further data exploration. The regulations of data changes among different activities were compared. Each activity group incorporated three types of visual data: the triaxial acceleration curve, the triaxial attitude angle curve and the surface temperature curve (Figure 4).

When sitting still, all types of data did not show an obvious change, and there were only slight fluctuations within a certain range of fixed values. This kind of activity signal can be judged with the naked eye and is easy to distinguish. However, when walking, walking upstairs and walking downstairs, all kinds of data showed periodic changes, which were caused by the arm swinging following a certain rule. In addition, results demonstrated that the body surface temperature in various moving activities was lower than that in the static state, and the main reasons for the temperature drop could be inferred as follows: (1) the data for the three moving activities (walking upstairs and downstairs) were collected outdoors, and the outdoor temperature was lower than the indoor temperature during the experiment, which had a certain influence on the measurement results; (2) in the moving activities, the participants’ arm swinging led to an increase of the air flow rate, and the acceleration of arm heat dissipation caused a decrease in the body surface temperature; (3) different somatic functions of the 10 participants also causes the different body temperature measurement results.

In comparison, the moving activities are difficult to differentiate intuitively to sitting, especially data for walking upstairs and downstairs. For the purpose of classifying activities more accurately, data preprocessing and different learning algorithms were used for further analysis and discussion to achieve more efficient and accurate human activity recognition, rather than simply judging the categories by setting the threshold value. Some ML and DL algorithms were used to train and test the activity dataset and then establish a more generalized learning model.

## 3. Activity Recognition Algorithm

Human activity recognition algorithm procedures include the standardization and preprocessing of the raw activity data, utilizing various machine learning and deep learning algorithms to identify human activities. The accuracy of all selected classifiers was compared and evaluated, and the main algorithm of this experiment is discussed in more detail below. In addition, the positive impact of including temperature data on the accuracy of human activity recognition was further verified by learning algorithms. The process of the proposed activity recognition algorithm was as follows.

### 3.1. Activity Data Collection

Ten healthy adults were assisted in wearing the wearable sensor with the same setup method. The HAR system gathered data at a rate of 1 s and read about 15 data points, with 50 to 60 s for one cycle, to obtain the acceleration, angular velocity and temperature data to complete the data collection process for sitting, walking, walking upstairs and walking downstairs activities. A total of 31,713 entries were obtained as the input of the raw dataset.

### 3.2. Signal Standardization and Data Stacking

A common approach of data pre-processing is standardization, which means centering data according to the mean (μ) and then scaling it on the basis of the standard deviation (δ). Then, data are obtained that follow a normal distribution with a mean of 0 and variance of 1, which is also called Z-score normalization [18]. The formula is as follows:(1)$$x^{*}=\frac{x-\mu }{\sigma }$$

Data stacking and resampling were performed for the standardized dataset according to the overlapping methods with different values (0%, 10%, 20%, 30%, 50%, 70%, 90%), and groups of 10 samples were taken as an instance. After feature engineering, the processed dataset was imported into the chosen learning algorithms, and the training set and the testing set were divided, accounting for 75% and 25% of the data, respectively. A comparison of the algorithms’ accuracies under different proportions was conducted, and the sliding window with a 50% overlap was attached great importance.

### 3.3. Learning Algorithm Accuracy

In order to screen the most suitable learning model for this activity recognition experiment on the basis of the sample dataset, some classical machine learning and deep learning algorithms were selected for data training and testing in this study, such as random forest (RF), support vector machine (SVM), K nearest neighbor (KNN), stochastic gradient descent (SGD), linear regression (LR), the Naive Bayes (NB), stacked denoising autoencoder (SDAE), extreme trees (ET) and deep forest (DF). As demonstrated in Table 3, the experiment compared the accuracy of various algorithms from different perspectives: (1) comparing the accuracy of different classifier types; (2) comparing of the effects of each algorithm in the six-dimensional dataset containing only inertial sensing data and the seven-dimensional dataset with temperature sensing data; and (3) comparing the performance of the seven-dimensional dataset containing temperature data with different data stacking modes. Table 3 demonstrates the previously mentioned comparison results of various classifiers used in this paper.

Initially, similar to many existing HAR experiments, only inertial data such as acceleration and gyroscope data were considered as inputs to the learning algorithm. However, the classification results were not satisfying. It can be seen that the better-performing classifiers random forest and extreme tree could merely achieve a 78% recognition rate, followed by support vector machine (74%), k-nearest neighbor (73%) and SDAE (75%). Several generalized linear classifiers, such as SGD and LR, did not perform well, and deep forest and Naive Bayes were not suitable either.

Therefore, we tried to add temperature sensing data to enrich the data dimension. The accuracy of all classifiers improved to different degrees (2–11%) as expected, except for SGD, which indicated that temperature data played an important role in improving the effectiveness of human activity recognition. In the case of no data stacking but considering body temperature, extreme forest was the most qualified algorithm, the accuracy of which was improved by 11% to 89%, followed by random forest with relatively better performance, the accuracy of which was improved by 10% to 88%, SVM and KNN achieved an 81% recognition rate, and the accuracy of the stacked denoising autoencoder could reach 78% after 5000 times of training. Data stacking and resampling were carried out on the seven-dimensional data set in the next step. Although the accuracy of each classifier did not change greatly before the overlapping reached 50%, it improved obviously when the percentage of data stacking went exceeded the bound.Random forest was the best performing learning model with an accuracy rate of 92%, followed by ET (90%) and KNN (83%), but the accuracy of the other algorithms did not improve significantly. With the increasing resampling rate, the accuracies of the main algorithms were further improved, and extra trees even reached the recognition rate of 96%. The basic logical architecture of learning algorithms is discussed in further detail below.

#### 3.3.1. Conventional Machine Learning

Machine learning involves the theoretical knowledge of interdisciplinary disciplines and is a science of artificial intelligence, using previous experience and data to optimize computer programs or improve algorithm performance [1,2,19]. Conventional machine learning models mentioned in this study are SVM, KNN, SGD, LR and NB.

Support vector machine: SVM aims to find an optimal hyperplane, and the largest interval hyperplane will classify samples by distinguishing between positive cases and other cases. The sample points closest to the hyperplane are called the support vector. The classification results of the SVM classifier are mainly affected by the kernel function, including the linear kernel, polynomial kernel and radial basis function kernel (RBF), which also called sigmoid kernels [20]. In this experiment, the SVM classification accuracy without temperature data reached 74%, and this number ascended to 81% after considering temperature. However, after data stacking, the accuracy dropped to 40%, which may have been caused by the increased model complexity.K-nearest neighbor: Compared with other classification methods, KNN has no obvious learning process, since it does not process data in the training stage but simply saves the obtained training samples. In addition, different k values may influence the classification results of KNN: a small k value may cause overfitting, while an overly large value may cause underfitting. The working principle of KNN is to classify data by measuring the distance between data, but when the data dimension is too high, it is difficult to calculate the distance between two samples, which results in large prediction deviation [21]. Different values of k were taken in the range of 1-100 to obtain the optimal solution in this experiment, as presented in Table 4.The results showed the highest accuracy when k was 7, reaching 81.26%.Stochastic gradient descent: SGD is commonly used to optimize learning algorithms; for example, building the loss function for the original model and finding the optimal parameter that minimizes the function value through the optimization algorithm. Each iteration uses a set of randomly shuffled samples to effectively reduce the parameter update cancellation phenomenon in small sample problems [22]. However, for the dataset used in this experiment, the performance of the SGD algorithm is very unsatisfactory, with an accuracy of less than 50% before data processing.Logistic regression: LR aims to organize samples of different categories distributed on both sides of the straight line as far as possible. To the best of our knowledge regarding the logistic regression function (sigmoid function), its output of a large range of numbers can be compressed within the interval of [0,1]. The maximum likelihood method is often used to estimate the parameters of LR, which is equivalent to the minimum likelihood loss function [23]. Likewise, the LR algorithm also performed poorly in this experiment, with only a 56% recognition rate.Naive Bayes classifier: The core concept of NB is to assume that the components of all vectors are independent of each other, but this also makes it unsuitable for problems with a large number of attributes or a large correlation between attributes [24]. The classification process of NB is divided into two main stages. The first stage is the learning, in which the classifier is constructed from sample data. The second stage is the reasoning, including calculating the conditional probability of nodes and classifying data. The accuracy of NB on the sample data set was only 61% and thus hardly required further consideration.

In brief, with the exception of the KNN algorithm, several traditional machine learning algorithms showed a low accuracy in human behavior recognition experiments. Therefore, we considered some deep learning algorithms for further research.

#### 3.3.2. Deep Learning

Deep learning is a research direction in the field of ML that accomplishes more complex classification tasks through feature learning [25,26,27]. DL emphasizes the depth of the model structure and clarifies the importance of feature learning. The SDAE algorithm is one of the deep learning algorithms.

Stacked denoising autoencoder: SDAE is a deep learning model. The auto-encoder (AE) is supposed to be introduced first, which is a self-monitoring algorithm. A simple AE model consists of an encoder and a decoder, and the data are input into the encoder and then into the decoder to obtain the final reconstructed data. Secondly, by adding noise to the input data, overfitting can be avoided, and thus the denoising autoencoder (DAE) is formed. DAE uses the data after adding noise for training, so the weight of the model contains less noise, thus improving the robustness and stability of the model. The SDAE model is used to stack multiple DAE models together to form a depth model. Each layer is independent for unsupervised training. The output of the first layer can be used as the input of the next layer, and the last layer is the softmax layer [28,29,30]. The SDAE model designed in this paper has three layers and was trained 100, 500, 1000, 1500 and 3000 times, respectively. The classification accuracy of the model can be improved by increasing the number of training times. However, it is found that when the training times reach a certain value, although the accuracy of the training set increases continuously, the accuracy of the testing set remains at about 79%. General speaking, although the training times increase, the testing accuracy of the model will not change when the training iterations reach a certain value. In order to balance the cost and accuracy, 500, 1000, 1500, 3000 and 5000 training iterations were performed. Consequently, when the number of training iterations reached 3000, the training accuracy was not greatly improved. A total of 3000 iterations is also a relatively moderate choice, which takes into account the training cost and classification accuracy; specific results are shown in Table 5.This chart shows that the classification accuracy was very poor when the number of training iterations was 100, which was of no significance. Therefore, the training iterations were increased to 500, and the result improved. When the repetitions were scaled up to 1500, the accuracy reached 75%. In order to find the best matching number, the training iterations were doubled to 3000, at which time the accuracy was only improved by 2%. In contrast, when the training times were increased from 100 to 500, the accuracy was improved by 10%, which proved that the accuracy of the SDAE model would not increase significantly with the increase of the training times when the training times were increased to 3000. When the training times reached 5000 times, even if the training set accuracy continued to increase, the accuracy of the testing set did not improve significantly, and finally stabilized at 78%.

The performance of SDAE, one of the deep learning algorithms, also failed to reach expectations. We decided to further attempt several integrated learning models to achieve better classification results.

#### 3.3.3. Ensemble Learning

Ensemble learning aims to integrate the results of all models by building multiple models on data, rather than a single machine learning algorithm. Ensemble algorithms aim to achieve better performance than a single model by considering the comprehensive modeling results, and pursuing accuracy, diversity and a better generalization ability. Ensemble learning models are generally classified into three categories: bagging, boosting and stacking [31]. The following part introduces several ensemble algorithms involved in this study: random forest, extra trees and deep forest.

Random forest: Random forest belongs to the bagging category (bootstrap-aggregating), which indicates the data are randomly extracted from the raw dataset and put back. Multiple decision trees are built through these subsets, and all classification voting results are integrated to ultimately obtain the average testing results of the classifier [32]. RF is simple and can be effectively applied to large datasets with good accuracy. In this study, the performance of random forest was outstanding, as expected, reaching the recognition rates of 88% and 89%, respectively, in the cases of 0% and 50% data stacking.Extra trees: ET is also called extremely randomized tree, which is also an DT-based ensemble learning algorithm, but it belongs to another random process rather than bagging. Compared with the ensemble methods represented by the RF, the main characteristics of ET are the way of constructing the trees in the forest and selecting the splitting points. There is no bagging process and it is unnecessary to use a bootstrap as a copy. Therefore, these features of ET weaken the correlation between the base estimators, simplify the process of node segmentation, reduce the complexity of model splitting, decrease the amount of computation, improve the training speed and form more diversified trees. When considering the bias–variance tradeoff in the model selection procedure, ET also has advantages, because its stronger random process can effectively reduce the variance. It also uses the whole training set to conclude each tree in the model, which can minimize the bias to a certain extent [33,34,35,36]. In practical application, the performance of extra trees is also related to the selection of parameters. In this experiment, the super parameters we selected were n_estimators = 550, random_state = 666, bootstrap = true, oob_score = true, and the accuracy was almost the same as RF or even higher. In the cases of 0% and 50% data stacking, the accuracy reached 89% and 90%, respectively. Parameters could be adjusted appropriately according to specific problems or by using cross-validation when necessary, which had a certain impact on the performance of the model.Deep forest: DF is a non-neural network deep tree model, which was originally proposed by professor Zhou Zhihua from Nanjing University in the study of an alternative to deep neural networks and is also known as the multi-grained cascade forest (gcForest). DF is a type of deep structure based on the logic of deep learning. Compared with the deep neural network, it is not only easier in terms of its theoretical analysis, but also simpler in terms of its parameter setting and training process, and it even shows more competitive performance on open datasets in some certain application domains. In the model training procedure, the deep neural network requires large-scale data, while DF can be trained on small-scale datasets, with relatively lower computational complexity [37].The general process of gcForest is composed of multi-grained scanning and the cascade forest. The first step is to preprocess the raw input features by using multi-grained scanning. In the second step, the feature vectors are input into the cascade forest for training, and the output of training data of each layer is used as the input of the next layer, and this is repeated continuously until the verification results converge. However, DF performed the worst of all the selected ensemble learning models in this experiment, with only a 71% recognition rate in the case of no data stacking.

#### 3.3.4. Further Attempt

In the processing of the existing raw dataset, ensemble learning models performed better, especially the random forest and extra trees. However, the results required some further attempts in order to find the suitable sample data type for DF. Taking the dataset that considered the temperature sensor data but no data stacking process as the standard, on the basis of the raw dataset, entries were randomly selected to form a new small sample dataset, and 50% (15,857 entries) and 10% (3171 entries) of the total quantity were retained, respectively. The variations of the algorithm accuracies for random forest, extra trees and deep forest are compared in Table 6.

Several conclusions can be drawn from different aspects: (1) when the total number of data samples was halved, the performance of RF and ET decreased, but the accuracy of DF increased; (2) when the data volume dropped from 100% to 10%, the accuracy of DF only decreased by 1%, while the other two algorithms decreased by 5% and 7%, respectively; (3) ET had the highest accuracy overall, but DF had a more robust characteristic for the small data samples. Therefore, the three ensemble learning algorithms have different advantages that need to be comprehensively considered in terms of the volume of the data itself and the degree of data stacking.

In conclusion, according to the accuracy value of recognition algorithms, only the conventional machine learning algorithm represented by KNN and several ensemble learning models performed better in this HAR experiment.

### 3.4. Algorithm Evaluation and Discussion

In addition to empirical evidence support, cross-validation and statistical testing methods are often selected to judge HAR classifier performance. For binary classification problems, classification results can be organized into a visual confusion matrix, in which the following values can be easily obtained and summarized [38].

True Positive (TP): the true category of the sample is positive, and the predicted result is also positive;True Negative (TN): the true category of the sample is negative, and the predicted result is also negative.;False Positive (FP): the true category of the sample is negative, but the model predicts it to be positive;False Negative (FN): the true category of the sample is positive, but the model predicts it to be negative.

Based on the accuracy results, KNN and ensemble algorithms were chosen for further comparison. In order to further evaluate the activity recognition results of several learning algorithms (RF, KNN, ET, DF) under the situation of no data stacking and 50% resampling, the evaluation metrics used in this paper were accuracy, precision, recall and *F*1 score.

Accuracy represents the proportion of correct prediction among all samples, defined as follows:(2)$$\;\mathrm{Accuracy}\;=\frac{TP+TN}{TP+TN+FP+FN}$$

Precision refers to the percentage of positive examples in which the prediction is correct:(3)$$\;\mathrm{Precision}\;=\frac{TP}{TP+FP}$$

Recall represents the proportion of positive cases that are correctly predicted among all samples:(4)$$\;\mathrm{Recall}\;=\frac{TP}{TP+FN}$$

The F1-score is the harmonic mean of accuracy and recall, which considers them to be equally important, and the formula is defined as follows:(5)$$F1\;\mathrm{score}\;=2\times \frac{\;\mathrm{Precision}\;\times \;\mathrm{Recall}\;}{\;\mathrm{Precision}\;+\;\mathrm{Recall}\;}$$

The activity sets in the study consisted of sitting, walking, walking upstairs and walking downstairs. The accuracy, precision, recall and F1 score of the confusion matrix are used in Table 7 to facilitate further comparative analysis of the classification performance of the four learning algorithms under different activity patterns. The reader is reminded that the discussion and evaluation are focused on the two perspectives of data without stacking and with 50% stacking respectively after data preprocessing. In addition, the time costs of different algorithms were taken into consideration simultaneously.

#### 3.4.1. Experimental Result without Data Stacking

All the data illustrated in the chart contain seven-dimensional features that include three-axis acceleration, three-axis angular velocity and body surface temperature. On the left side of the table are the results for each classifier without performing data stacking.

Evaluation metrics under the static state are generally higher than those under a dynamic state. When sitting, the recognition rate of the majority of the algorithms reaches 98% or 99%, which means that they can basically realize accurate activity recognition. Secondly, the error of both recall and F1 score for walking is smaller than that for walking upstairs and downstairs, except that the value of precision is not stable in different situations. For the comprehensive comparison of the three indicators under the three dynamic activities, RF and ET showed similar performance, followed by KNN, and DF was the worst. Likewise, the accuracy of the RF and DT algorithms is the same, reaching 89%, while KNN reached 83% and DF only obtained a 70% recognition rate; in particular, the prediction of upstairs and downstairs labels is relatively inaccurate. Regarding the time consumption of the algorithm, KNN took the shortest time, at less than 0.1 s, but it is not accurate enough. DF took the longest time, which is 41.6 s, which is related to the complex structure of depth, while RF took a shorter time than DT, at 4.5 s. Considering the evaluation criteria and running time comprehensively, RF can be seen to be most appropriate for this dataset. It can be observed from the values of the confusion matrix that walking upstairs and downstairs labels are prone to be confused. In other words, part of the walking labels were predicted as walking upstairs and downstairs, or vice versa, which may be caused by the process of manual labeling; the data of walking at the corner of stairs up and down were mixed into the real labels, resulting in errors in the classifier. In order to reduce such human errors, we considered using the method of data stacking in subsequent experiments. Ten entries were combined into one instance by using the sliding window; then, the data were divided into different proportions, and the learning model was retrained.

#### 3.4.2. Performance Enhancement with 50% Stacking

To solve the existing problem of the inaccurate classification of upstairs and downstairs labels caused by man-made errors after feature processing, in this paper, a data stacking approach with different proportions was used for further research, with stacking ratios of 10%, 20%, 30%, 50%, 70% and 90%, respectively. When the stacking degree reached 50%, each classifier had a relatively obvious improvement in accuracy. Although the algorithm accuracy under 70% and 90% is very high, the degree of stacking is also too high to consider. Therefore, this part mainly focuses on the different evaluation indicators of the four algorithms in the case of 50% stacking. According to the evaluation results of the algorithms on the right side of the chart, the indicators of each algorithm have improved to a certain extent, especially in the three dynamic states of walking, walking upstairs and walking downstairs. The accuracy of RF and DT was improved by 0.02% as well, reaching 91%. However, considering the running time, the RF took 2.38 s, which was shorter than the 4.05 s of DT.

## 4. Human-Centered Application in COVID-19

The COVID-19 is currently under control in some countries and regions, and many new application demands based on wearable sensing technology are also constantly emerging. However, the quality and efficiency of the follow-up long-term prevention and control efforts will be of greater concern. Extreme weather conditions and human activity states influence the accuracy of temperature and also affect the efficiency of body temperature monitoring. Particularly during the epidemic period, when online consumption has increased sharply, the demand for express delivery orders and takeout orders has been on the rise, and the temperature monitoring of service personnel on various platforms is also an issue of great concern. However, the constant switching activities of couriers and takeaway staff and the change of ambient temperature can easily lead to inaccurate measurements of body surface temperature. Therefore, the accurate and real-time monitoring of body temperature can bring consumers a more secure experience. In brief, strict screening and timely isolation can be used to respond effectively to epidemic prevention and control, so body temperature monitoring is a key step of daily epidemic prevention, and infrared thermometers play an important role as a common temperature measurement tool.

In the field of HAR, including the related work summarized before, current wearable sensing technology still largely relies on inertial sensing data, mainly gathering data from an accelerometer and gyroscope to recognize human activities, while rarely involve temperature data or consider temperature sensors. Nevertheless, in the context of COVID-19, body temperature has become a sensitive topic that cannot be ignored in daily activities, so this experiment has tried to include the measured body surface temperature data into the collected dataset. According to the experimental results, it can be found that the accuracy of the learning model is indeed improved after incorporating the temperature data simultaneously, and it also has a better generalization ability. Therefore, the body surface temperature data in this experiment are beneficial to human activity recognition algorithms, increase the data dimension and improve the accuracy of HAR technology. Moreover, participants’ body surface temperatures measured under the three dynamic activities were slightly lower than that in the static state. In order to discuss the temperature deviation more intuitively and specifically, the body surface temperature of the participants while sitting was taken as the reference dataset (Ts). The body surface temperature data while walking (Tw), walking upstairs (Tu) and walking downstairs (Td) were subtracted from the reference dataset, respectively, to obtain the quantitative temperature deviation data of different activities. Then, a statistical analysis was conducted on sample data to output the three normal distribution diagrams (Figure 5).

Visualized data show that the three temperature curves under the dynamic activities present a normal distribution, and the expectations and variance are different. Table 8 displays the specific values. It can be seen that the body surface temperature while walking is on average 1.15 ${}^{{}^{\circ}}$C lower than that while sitting, with a variance of 0.82. Similarly, the temperature difference expectation of walking upstairs is −1.17 with a variance of 0.78, while the expectation of walking downstairs is −0.95 with a variance of 0.88. Therefore, the results were linked to the actual application scenarios to avoid the omission of feverous patients and the potential infectious diseases caused by the change of body surface temperature in dynamic activities during COVID-19 prevention and control. It can be more convenient to calibrate the body temperature by analyzing and inferring the tested human activity, then referring the specific expectations and mean values of different activities, such as those shown in Figure 6. In this way, the quality of epidemic prevention work can be improved to a certain extent and the problems that may be faced under the normalization of COVID-19 will be handled better.

## 5. Conclusions

This paper considers wearable sensors and learning algorithms to realize HAR, and a temperature sensing module is added to further explore the relationship between human body surface temperature and activity transformation in the context of COVID-19. Ten volunteers were selected from different occupations, heights and weights, covering different age groups to ensure that the selected learning model is generalizable enough to adapt better to future new users. After preprocessing the raw database, ML and DL models were able to identify the four activities—sitting, walking and walking upstairs and downstairs. In addition, the performances of the selected algorithms were overall improved after taking temperature sensing data into consideration. The ensemble models—ET and RF—performed better in terms of the algorithm accuracy, the evaluation metrics report and the algorithm running time. With the data stacking and resampling processes, they were able to reach 90% and 92% recognition rates, respectively, which are slightly improved values compared to many previous related works. The most significant result was that people’s body temperatures during dynamic patterns were lower than during sitting, which is related to temperature monitoring during COVID-19. Only a 1–2 ${}^{{}^{\circ}}$C temperature error may lead to the omission of potential fever sufferers and affect the accuracy and efficiency of epidemic prevention and control workflow. In this case, the temperature adjusting criterion provided in this paper based on statistical data analysis can provide medical care personnel with an intuitive reference for future COVID-19 normalization prevention.

## Figures and Tables

**Figure 1 sensors-21-07540-f001:**
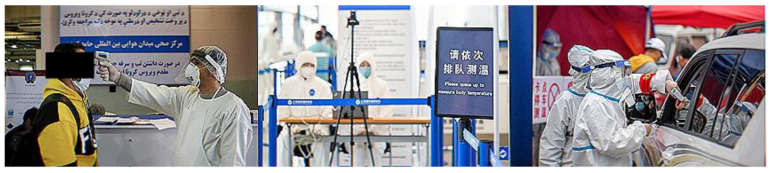
Temperature measurement in public places for COVID-19 regular prevention.

**Figure 2 sensors-21-07540-f002:**
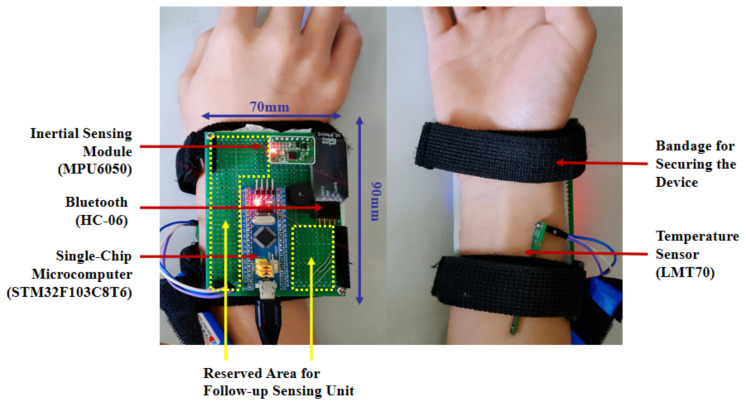
Hardware structure of the wearable device.

**Figure 3 sensors-21-07540-f003:**
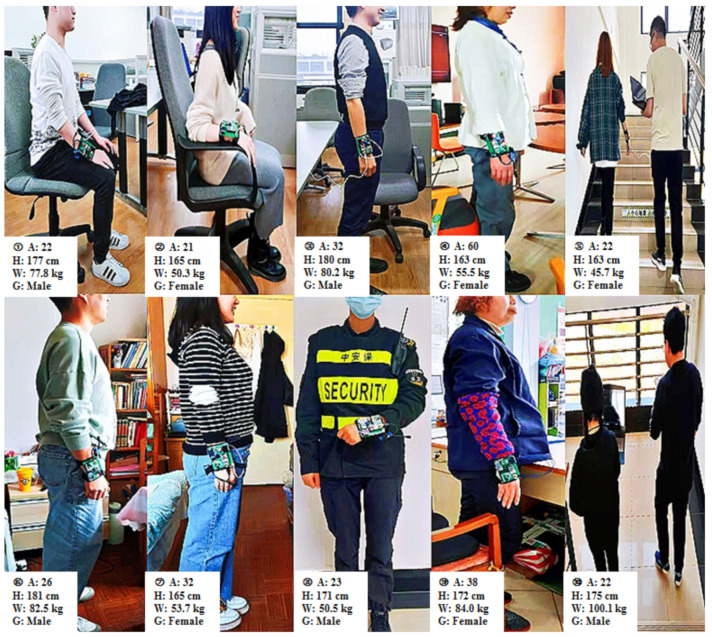
The 10 participants wore wearable sensors to conduct different activity recognition experiments. (A, H, W and G are the abbreviations of age, height, weight and gender).

**Figure 4 sensors-21-07540-f004:**
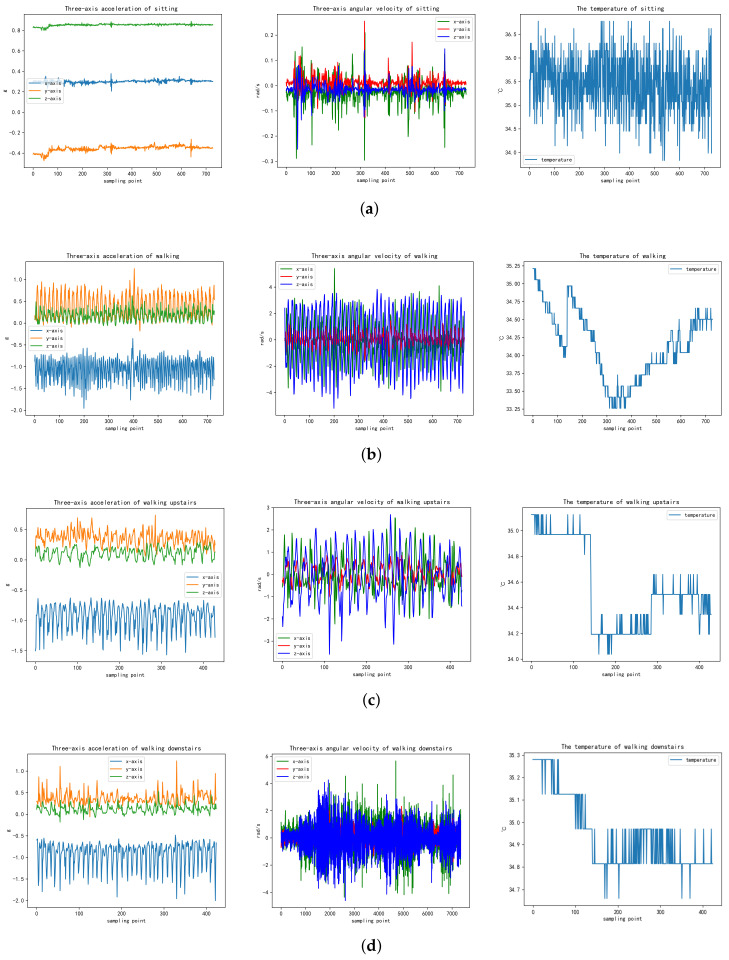
Visualized sensing signal curves for four activity patterns (sitting, walking, walking upstairs and walking downstairs). (**a**) Sitting with acceleration, velocity and temperature sensing signal curves. (**b**) Walking with acceleration, velocity and temperature sensing signal curves. (**c**) Walking upstairs with acceleration, velocity and temperature sensing signal curves. (**d**) Walking downstairs with acceleration, velocity and temperature sensing signal curves.

**Figure 5 sensors-21-07540-f005:**
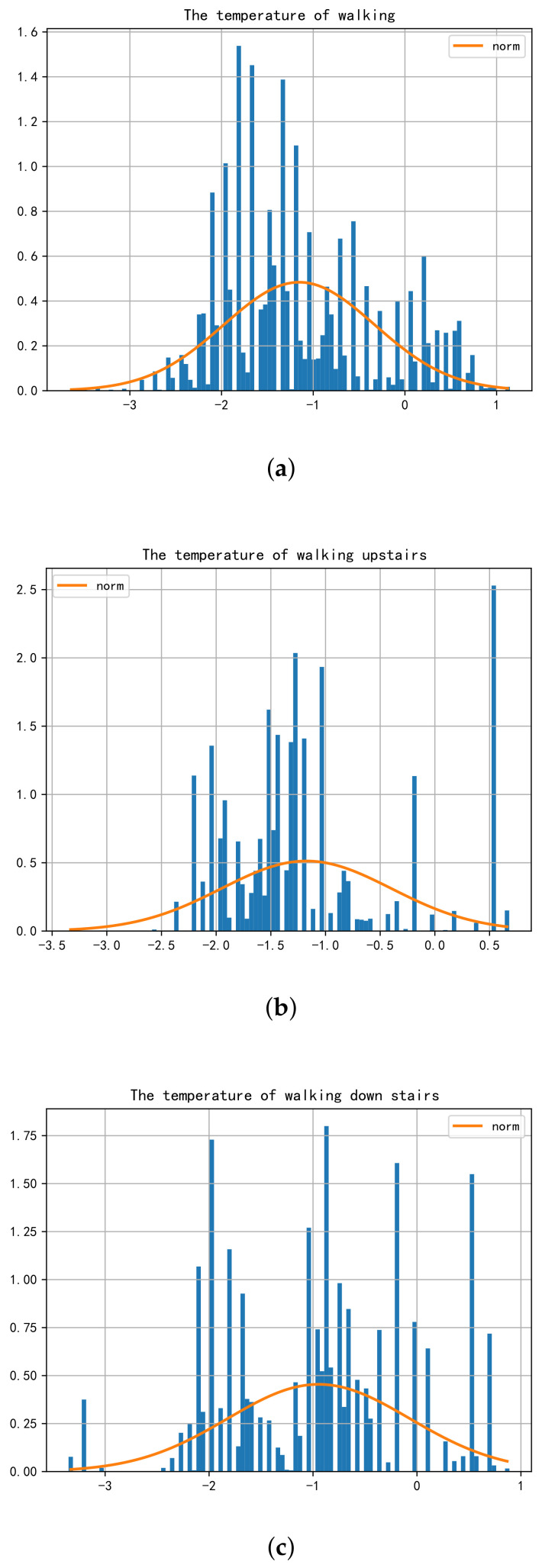
Normal distribution curves of body surface temperature error under dynamic activities. (**a**) The temperature while walking. (**b**) The temperature while walking upstairs. (**c**) The temperature while walking downstairs.

**Figure 6 sensors-21-07540-f006:**
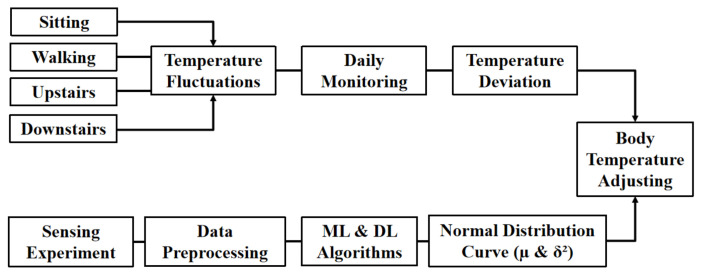
General workflow for temperature monitoring and adjusting.

**Table 1 sensors-21-07540-t001:** Hardware module.

Hardware Component	Model Type
Single-Chip Microcomputer	STM32F103C8T6
Inertial Sensing Module	MPU6050
Temperature Sensor	LMT70
Bluetooth Module	HC-06
Power Supply Module	5V and 1500 mAh dry battery

**Table 2 sensors-21-07540-t002:** Physical parameters of the 10 participants.

ID No.	Occupation	Age	Height (cm)	Weight (kg)	Gender
1	Student	22	177	77.8	Male
2	Student	21	165	50.3	Female
3	Professor	32	180	80.2	Male
4	Cleaner	60	163	55.5	Female
5	Student	22	163	45.7	Female
6	Police	26	181	82.5	Male
7	Accountant	32	165	53.7	Female
8	Security	23	171	50.5	Male
9	Staff	38	172	84.0	Female
10	Student	22	175	100.1	Male

**Table 3 sensors-21-07540-t003:** Accuracy of activity recognition algorithms.

		Accuracy				Data	Stacking		
**Learning Algorithms**	**Classifiers**	**(No Temperature)**	**0%**	**10%**	**20%**	**30%**	**50%**	**70%**	**90%**
	SVM	74%	81%	40%	43%	43%	48%	47%	-%
	KNN	73%	81%	80%	80%	83%	83%	85%	93%
Conventional Machine Learning	SGD	50%	47%	52%	55%	53%	55%	50%	54%
	LR	54%	56%	58%	56%	58%	56%	55%	58%
	NB	59%	61%	63%	61%	62%	62%	62%	59%
Deep Learning	SDAE	75%	77%	69%	72%	74%	75%	75%	86%
	RF	**78%**	**88%**	89%	89%	89%	**89%**	92%	91%
Ensemble Learning	ET	**78%**	**89%**	88%	89%	89%	**90%**	92%	96%
	DF	65%	71%	77%	76%	74%	75%	75%	79%

**Table 4 sensors-21-07540-t004:** Classifier accuracy corresponding to different K values.

K Value	Accuracy	K Value	Accuracy
1	80.05%	13	80.89%
2	78.74%	14	80.87%
3	81.22%	15	80.64%
4	80.82%	16	80.42%
5	80.93%	17	80.24%
6	80.97%	18	80.18%
7	81.26%	19	80.16%
8	80.90%	20	79.87%
9	80.75%	30	78.61%
10	80.93%	50	76.81%
11	80.74%	80	75.18%
12	80.94%	100	74.22%

**Table 5 sensors-21-07540-t005:** Accuracy of SDAE classifier with different training times.

Frequency of Training	100	500	1500	3000	5000
Training set accuracy	65%	73%	80%	85%	88%
Testing set accuracy	61%	71%	75%	77.5%	78%

**Table 6 sensors-21-07540-t006:** Accuracy comparison of ensemble algorithms.

Processing Mode	No. of Dataset	Accuracy		
		RF	ET	DF
Raw dataset	31,713	88%	89%	71%
50% of the total	15,857	87%	86%	73%
10% of the total	3171	83%	82%	70%

**Table 7 sensors-21-07540-t007:** Algorithms’ evaluation metrics for activity recognition.

Algorithm	ACT	0% Precision	Data Recall	Stacking F1 Score	Accuracy	Running Time	50% Precision	Data Recall	Stacking F1 Score	Accuracy	Running Time
	Sitting	0.99	0.99	0.99			0.99	0.98	0.99		
RF	Walking	0.86	0.90	0.88	0.89	4.50 s	0.86	0.92	0.89	0.91	2.38 s
	Upstairs	0.86	0.81	0.83			0.92	0.83	0.87		
	Downstairs	0.86	0.86	0.86			0.89	0.90	0.90		
	Sitting	0.98	0.99	0.99			0.96	0.98	0.97		
KNN	Walking	0.77	0.82	0.80	0.83	0.07 s	0.75	0.86	0.80	0.83	0.01 s
	Upstairs	0.73	0.73	0.73			0.77	0.69	0.73		
	Downstairs	0.81	0.72	0.76			0.86	0.77	0.81		
	Sitting	0.99	0.99	0.99			0.99	0.99	0.99		
ET	Walking	0.84	0.91	0.87	0.89	10.66 s	0.83	0.93	0.88	0.91	4.05 s
	Upstairs	0.88	0.78	0.83			0.92	0.79	0.85		
	Downstairs	0.86	0.85	0.85			0.91	0.89	0.90		
	Sitting	0.98	0.98	0.98			0.99	0.97	0.98		
DF	Walking	0.58	0.78	0.67	0.70	41.56 s	0.59	0.67	0.63	0.69	6.81 s
	Upstairs	0.61	0.44	0.51			0.57	0.49	0.53		
	Downstairs	0.65	0.53	0.59			0.61	0.60	0.61		

**Table 8 sensors-21-07540-t008:** Values of expectation and variance in the normal distribution curve.

Temperature Difference ( ${}^{{}^{\circ}}$C)	Expectation (μ)	Variance ($\delta^{2}$)
Walking (Tw-Ts)	−1.15 ${}^{{}^{\circ}}$C	0.82 ${}^{{}^{\circ}}$C
Upstairs (Tu-Ts)	−1.17 ${}^{{}^{\circ}}$C	0.78 ${}^{{}^{\circ}}$C
Downstairs (Td-Ts)	−0.95 ${}^{{}^{\circ}}$C	0.88 ${}^{{}^{\circ}}$C

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
