# Peer review of "Body Temperature Monitoring for Regular COVID-19 Prevention Based on Human Daily Activity Recognition"

_sensors, 2021, doi:10.3390/s21227540_

Round 1
Reviewer 1 Report
This paper presents an approach for adjusting body temperature using human activity recognition approach. Although at first the research goal of this paper appears significantly interesting, the paper has several weaknesses.
1) The number of subjects is significantly small and the ages of the subjects are not representative of the population (9 subjects between 21-39 and 1 subject 60 years old)
2) The authors present too many technical details.
3) The authors present too many theoretical details regarding the classifiers used and the metrics used for evaluation
4) It is not clear whether focus is given on HAR using inertial sensors or body temperature adjustment. If the goal is the former, the work has no novelty and the number of classes is significantly small, while classes are not representative of daily activities. If the goal is the latter, then all the aforementioned details are unnecessary, yet the evaluation is not appropriate
5) The evaluation does take into account issues such as discomfort and usability
6) The evaluation of the case study is not appropriate. Results as shown in Table 8 are not clear of the actual performance
7) There are many typos and several parts that need rephrasing
Reviewer 2 Report
This paper is based on wearable sensors and learning algorithms to complete human activity recognition with the support of temperature sensors. The review provides a major concern on the novelty/significance of the work. There are many implementations of inertial sensors with STM32. Adding a temperature chip or a BLE device is not new. Also, the authors implemented ML/DL methods on HAR, which has been explored in much previous research. A simple comparison might not be enough.
This paper could fit as a conference paper; for an MDPI Journal, significant novelty is required.
Reviewer 3 Report
Review of the article „Body Temperature Monitoring for COVID-Regular Prevention based on Human Daily Activity Recognition“ authors: Lei Zhang, Yanjin Zhu, Mingliang Jiang, Yuchen Wu, Kailian Deng, Qin Ni
Shortcomings of the article:
In figures 4 and 5 need marked graph a and graph b and them explanations and the comments are difficult to read. The everything needs to be increased.
All researches are performed according to the established methodology, need provide the methodology of these experiments.
Why all tables are in different format?
What is the reliability of the results obtained? The human temperature can rise for a variety of reasons, so temperature measurement is not an objective parameter. How to distinguish that it is e.g. not the flu because the symptoms can be very similar.
In the conclusions must clearly show what problems the researchers have solved and how much to get results are better than the results of other researches. The conclusions should be clear and concise with the numerical values provided to support and justify the results obtained. The presented conclusions are not informative. Conclusions need to be rewritten.
This work is more like a report but not a scientific article.
Round 2
Reviewer 2 Report
The authors addressed the previous comments
Reviewer 3 Report
Accept in present form.